# Using an Electronic Goniometer to Assess the Influence of Single-Application Kinesiology Taping on Unstable Shoulder Proprioception and Function

**DOI:** 10.3390/s25072326

**Published:** 2025-04-06

**Authors:** Ewa Bręborowicz, Izabela Olczak, Przemysław Lubiatowski, Piotr Ogrodowicz, Marta Ślęzak, Maciej Bręborowicz, Leszek Romanowski

**Affiliations:** Orthopaedics, Traumatology and Hand Surgery Department, Poznan University of Medical Sciences, 28 Czerwca 1956 no 135/147, 61-545 Poznan, Poland; ewabreborowicz@ump.edu.pl (E.B.); iza.majewska@wp.pl (I.O.); p.lubiatowski@rehasport.pl (P.L.); mslezak@orsk.pl (M.Ś.); mbrebor@ump.edu.pl (M.B.)

**Keywords:** propriometer, electronic goniometer, glenohumeral joint, joint position sense, shoulder instability, kinesiology taping, Western Ontario instability shoulder index

## Abstract

**Background:** Glenohumeral joint instability is associated with a proprioception deficit. Joint position sense can be improved through targeted exercises and kinesiology taping (KT). While previous studies have examined the effects of KT on proprioception, most have focused on the knee joint, with limited research on unstable shoulder joints. Most studies have used commonly available equipment (e.g., the Biodex system). An electronic goniometer, the “Propriometer”, is a useful tool for assessing proprioception in shoulder joint instability; however, its application in evaluating the effects of KT on shoulder proprioception remains unexplored. This study aimed to (1) assess the usability of the Propriometer for evaluating the effects of KT on unstable shoulders and (2) determine the impact of a single KT application on joint position sense and limb function in individuals with anterior, post-traumatic shoulder joint instability. **Methods and Materials:** The study included 30 individuals with anterior, unilateral, post-traumatic shoulder joint instability (8 women, 22 men, mean age 26 years). A control group consisted of 35 healthy volunteers (9 women, 26 men, mean age 24 years). Proprioception assessment (active joint position reproduction evaluation) was performed in both groups using the Propriometer, which measures joint position in real time with an accuracy of 0.1° across all axes. The study methodology was validated and used to examine shoulder proprioception. The current study focused on assessing the effects of KT, which had not been previously tested with this device Assessments were conducted before KT application and three days’ post-application. Additionally, patients completed the Western Ontario Shoulder Instability Index (WOSI) self-assessment questionnaire before and three days after the therapy. **Results:** Results of the mean joint position reproduction error indicate a proprioceptive deficit in patients with shoulder joint instability. However, the analyzed KT application did not show a significant change in the magnitude of the active joint position reproduction error. Conversely, KT therapy significantly improved patients’ subjective assessment of shoulder function and stability as measured by the WOSI. **Conclusions:** The Propriometer goniometer and testing methodology are effective tools for assessing the impact of KT on proprioception in shoulder instability. While KT application did not significantly influence shoulder proprioception, it did improve patients’ perceived joint stability and function.

## 1. Introduction

The kinesiology taping (KT) method is increasingly used in the rehabilitation and prevention of shoulder disorders to reduce pain [1,2,3,4], enhance joint stability [4], and influence joint proprioception [5,6]. In post-traumatic anterior shoulder instability, significant proprioception deficits have been observed in joint position sense testing, linked to damage in the joint capsule, ligaments, mechanoreceptors, and free nerve endings [6,7,8,9,10,11]. Shoulder proprioception can be improved through targeted exercise programs, such as the Proprioceptive Neuromuscular Facilitation method. While surgical intervention has been shown to restore proprioception deficits in post-traumatic shoulder instability, there are limited data on the effects of KT on this function [12]. Previous studies have only evaluated the impact of KT on shoulder position sense in healthy individuals [6].

The KT therapeutic method is based on the anatomy and mechanics of the fascia [4], particularly its ability to transfer tension between the fascia and muscle fibers, and vice versa [13,14]. The improvement in joint proprioception may result from the positional sense provided by the skin, the body’s largest sensory organ [4]. Simoneau et al. suggest that taping along the joint provides significant sensory stimulation, leading to improved motor control [15]. Other researchers link the increase in proprioceptive stimulation to heightened activation of skin mechanoreceptors, which enhances motor unit recruitment in muscles [16]. The greater the tape stretch, the more significant the skin receptor stimulation [4].

The application used in our study aimed to improve proprioception and increase shoulder joint stability. Previous studies have examined the effect of KT on shoulder proprioception in healthy individuals, patients with subacromial impingement syndrome, and those with rotator cuff tendinopathy. However, there are a lack of studies investigating this relationship in shoulder joint instability [17,18,19,20]. The electronic goniometer, the “Propriometer”, and its testing methodology have been validated and previously used in proprioception studies [21]. While the Propriometer has been used to assess proprioception in shoulder instability, no studies have explored the effects of KT on an unstable shoulder using this device [11,21,22].

The aim of this study was to assess the usability of the Propriometer goniometer for examining the effect of KT on an unstable shoulder and evaluate the impact of a single KT application on joint position sense and limb function in anterior, post-traumatic glenohumeral joint instability

## 2. Materials and Methods

The study group consisted of 30 patients, aged 18–44 years (mean age 26 years, SD 6). The group included 8 women (mean age 27.1 years) and 22 men (mean age 26.9 years). All patients were diagnosed with unilateral, post-traumatic, anterior shoulder instability based on their medical history, clinical examination, and imaging studies. The control group consisted of 35 people (9 women and 26 men), aged 21–31 years (mean age 24 years, SD 2). In the patient group, two measures were taken: shoulder joint position sense and shoulder function in everyday activities, assessed using the Western Ontario Shoulder Instability Index (WOSI). Both tests were performed before and three days after a single application of KT. In the control group, the joint position sense test was also conducted before and three days after KT application. The three-day observation period was chosen because the elastic polymer properties of the tape decrease after this time [4]. A single application was selected, consistent with other studies examining the effects of KT [6,16,18,19,20].

For the study, KT manufactured by Nitto Denko Corporation (Osaka, Japan) was used. The KT application (Figure 1) combined three techniques: the ligament technique (the tape was applied with maximum stretch over the acromioclavicular joint), the modified muscular technique (on the deltoid muscle), and the functional technique (from the shoulder process to the distal attachment of the deltoid muscle, the tape was applied starting in a 90° elevated position, with gradual lowering of the limb as it was adhered).

The Propriometer was used to assess joint position sense, measured in degrees of joint position reproduction angle. The device was designed and manufactured by Progress Company (Ostrow Wielkopolski, Poland) specifically for studying shoulder proprioception. The Propriometer is an accelerometer that measures the range of motion and reads the joint’s position in space in real time, with an accuracy of 0.1° across all axes. The accelerometer has a chamber of gas with a heating element in the center and four temperature sensors around its edge. 

When the accelerometer is level, the hot gas pocket rises to the top center of the chamber, and all the sensors will measure the same temperature. By tilting the accelerometer, the hot gas will collect closer to some of the temperature sensors. By comparing the sensor temperatures, both static acceleration (gravity and tilt) and dynamic acceleration can be detected. The accelerometer converts the temperature measurements into signals (pulse durations) that are easy for microcontrollers to measure and decipher (Figure 2). The position-sensing transducer was fixed to the arm or forearm. The data were processed by dedicated software, which displayed the target joint angle set by the examiner and the reproduced shoulder angle measured by the patient. The system then calculated the joint position reproduction error.

The study methodology was validated and applied in studies of joint position sensation in both healthy and unstable shoulder and elbow joints [11,21,22,23]. The study of joint position sense was based on active-assisted demonstration of the reference position, and active reproduction of this position is considered to be the most sensitive and repeatable technique for testing this function [11,21,22,23]. In our study, the patient had their eyes covered. The examiner demonstrated the target angle by moving the limb, and the patient then actively reproduced this angle. Four shoulder joint movements were studied: abduction and flexion in a sitting position at angles of 60°, 90°, and 120° (Figure 3), and internal and external rotation in a supine position at angles of 30°, 45°, and 60°. The measurement result is the error of active reproduction of the joint position (EARJP). A lower EARJP value indicates better joint position sense.

Shoulder joint function was evaluated using the Western Ontario Shoulder Instability Index (WOSI). The survey consists of 21 questions divided into four categories: physical symptoms, sports and recreation, lifestyle, and emotions. The results are expressed as a percentage [24,25].

The study was approved by the ethical committee, and all participants provided signed written informed consent. Statistical analysis was performed using Statistica^®^10 software (StatSoft). Compliance with the normal distribution was checked using the Shapiro–Wilk test. For variables consistent with the normal distribution, the parametric Student’s *t*-test was used to assess significance. When the distribution was not normal, non-parametric tests were applied (Wilcoxon signed-rank test, Mann–Whitney *U* test). The level of statistical significance was defined as *p* < 0.05.

## 3. Results

### 3.1. Study Group

After KT therapy for an unstable joint, the EARJP value showed no statistically significant change compared to the EARJP value before taping (Table 1).

### 3.2. Control Group

The EARJP in the taped joint increased at most of the tested angles after applying taping compared to the pre-taping test (Table 2), but without statistical significance.

### 3.3. Study Group vs. the Control Group Before KT

The EARJP results for all angular settings in abduction and flexion in the unstable joint group were higher compared to the control group (Table 1 and Table 2). Statistically significant differences were observed in abduction at 90° and flexion at 60° (Table 3). In extreme external and internal rotation at 60°, the EARJP values in the unstable joint group before KT were lower than in the control group before KT. Statistically significant differences were found in external rotation at 45° and 60° and internal rotation at 30° and 60° (Table 3).

### 3.4. Study Group vs. the Control Group After KT

The EARJP values for all angular settings in abduction and flexion in the unstable joint group after KT were higher compared to the control group after KT (Table 1 and Table 2). Statistically significant differences were observed in abduction at 60° and 90° (Table 3). In all external rotation settings and internal rotation at 60°, the EARJP results in the unstable shoulder group after KT were lower than in the control group after KT (Table 1 and Table 2). Statistically significant differences were found in external rotation at 45° and 60° and internal rotation at 30° (Table 3).

### 3.5. Functional Assessment

All participants from the instability group (n = 30) completed a functional self-assessment using the WOSI questionnaire. The overall score from the survey was an average of 45% before KT therapy and 52% after therapy (Figure 4). The results improved in each question group, with all differences being were statistically significant (Figure 4).

## 4. Discussion

This study used an electronic goniometer, which measures the joint position angle with high precision. Patients were shown a target angle that they had to reproduce independently. The study measured the error in reproducing the joint position. During the examination, patients were in a comfortable sitting or lying position. The study methodology was made clear to the patients, and the lightweight sensor on the upper limb, made the examination easy to perform. The study did not cause fatigue or strain on the shoulder.

The results obtained in our study, as well as those from other studies, indicate deep sensation deficits in people with post-traumatic shoulder joint instability, which may be associated with damage to the capsule–ligament apparatus, mechanoreceptors, and free nerve endings [6,7,8,9,10]. The EARJP test for an unstable joint in the study group during abduction and flexion was higher compared to the control group for all angular settings, with statistical significance at abduction 90° (*p* = 0.0024) and flexion 60° (*p* = 0.0077). The position of abduction at 90° with external rotation is the most commonly reported cause of apprehension of dislocation; hence, a significant disturbance of joint position sensation may occur in this position. In the study by Jerosch et al., significant differences were found only at the maximum angles studied [9]. In our study, significant differences were observed in lower and mid-positions. In these positions, the joint capsule and ligaments are less involved in maintaining joint stability, and therefore, less information about the position of the limb comes from the capsule–ligament receptors, which may explain the higher EARJP results. In all external rotation settings, EARJP results were lower in the unstable group compared to the control group (ER 30°: 3.18 vs. 3.29 *p* = 0.7873; ER 45°: 2.10 vs. 3.11; *p* = 0.0227, ER 60°: 1.06 vs. 3.16, *p* = 0.0001). The statistical significance for these measurements is high. These findings may explain the occurrence of a positive apprehension test in individuals with unstable shoulders during external rotation, which makes patients more focused on this movement. In addition, EARJP decreases as the angle of external rotation increases, indicating better motion control at the extreme range of motion. This can be explained by increasing tension in the joint capsule as the range of motion increases, which activates more capsule receptors. A similar relationship was found by Blasier et al., showing better proprioception in external rotation compared to internal rotation, and better control of motion as it approached the extreme range of motion [26]. A similar relationship in the movements of lifting the upper limb was shown by Lubiatowski et al. In flexion and abduction at 60°, the EARJP values were higher than in the settings of 90° and 120° [11].

The selected KT application was intended to increase the shoulder joint stability and thus improve the joint’s deep sensation and the limb’s function in everyday activities. There have only been a few papers published on the influence of KT on glenohumeral joint proprioception [5,6,18,19,20]. The EARJP values measured in the study and control groups did not statistically significantly change after using KT. After KT application in the unstable shoulder group, the EARJP values decreased in 6 out of 12 examined angular positions, while in the healthy group, the values decreased in 4 positions. The analysis by Lin et al. of the error in reproducing the position of the shoulder joint after scapular taping in healthy people using an elastic patch showed a significant reduction in this parameter [6]. Scapular taping in healthy people using the McConell method with an inelastic patch showed no significant changes in the error in reproducing the joint position [18]. Callaghan’s research confirms the relationship between the use of the McConnel patch on the patella in healthy people and the change in the activity of specific brain areas depending on the level of blood oxygenation (BOLD) using magnetic resonance imaging [27]. Under the influence of the applied patch, there was an increase in activity in the medial part of the supplementary motor cortex, in the basal ganglia, in the thalamus, and in the medial part of the primary sensory cortex, and a decrease in activity in the lateral part of the primary sensory cortex [27]. These changes may indicate that the shift in deep sensation is related to the stimulation of mechanoreceptors and nervous-system activation, rather than supporting joint mechanics [4,27]. The study with patellar taping demonstrated that taping modulates brain activity in several areas of the brain during a proprioception knee movement task. This decreased BOLD response in the lateral primary sensory cortex was associated with less activity for the knee proprioception task with taping [27]. We can only suppose that a similar impact could be observed in an MRI examination during shoulder taping. The limited studies on this topic are not sufficient to clearly demonstrate the differences and real impact of kinesiology taping on a specific joint. Based on our and other authors’ results, it seems that despite the lack of statistical significance, the EARJP values are decreasing, suggesting that kinesiology taping is worth applying.

We assumed that increasing the stability of the shoulder joint should translate into improved extremity function in everyday activities. The overall result of the shoulder function test using the WOSI questionnaire after KT therapy increased from 45% to 52%; *p* = 0.0001. The WOSI questionnaire is recommended by many authors as a reliable and repeatable tool for assessing shoulder function in shoulder joint instability [24,25]. Improvement of shoulder function after applying the KT in terms of the patient’s subjective feeling may result from analgesia, supporting muscle work, and stabilizing effects. According to Alexander et al., the application of the tape running along the muscle can affect the shortening of muscle fibers, thus causing a decrease in the flow of afferent stimuli from the neuromuscular spindles, which further leads to a smaller recruitment of motor neurons of the anterior vertebral cord. The reduction in the recruitment of neurons located in the anterior horns of the spinal cord in the EMG study is manifested by a decrease in the amplitude of the H reflex response [2]. Such a relationship indicates an inhibitory effect of the tape, causing a decrease in muscle tone and limiting the formation of myofascial trigger points, which results in a reduction in pain [3]. However, there are no clear results confirming the actual analgesic effect of the KT method. Studies on the effects of KT on shoulder pain indicate only a short-term analgesic effect or its complete absence [17,28]. Hsu et al. reported that the effect of increased muscle activity under the influence of KT therapy is associated with the mechanism of paving impulses, through temporal and spatial summation of subliminal impulses, which leads to an increase in muscle excitability [28]. Hsu showed the supporting effect of elastic tape on the activity of the trapezius muscle in athletes with subacromial impingement syndrome [28]. It can be assumed that the mentioned mechanisms of the analgesic effect and supporting muscle work may translate into a general improvement in limb function.

The results of this study on the assessment of shoulder function after using dynamic taping therapy seem valuable and bring practical solutions for physiotherapists and orthopedists. In the future, it is worth expanding this study to analyze the combination of KT therapy with exercise. Studies published on disorders other than shoulder instability have shown that a combination of exercises with KT resulted in good functional effect [1,29]. Another limitation of our study is the short observation time; in the future, it would be beneficial to extend the observation period.

## 5. Conclusions

The Propriometer goniometer is a useful tool for assessing the impact of kinesiology taping on post-traumatic anterior shoulder instability. This condition affects joint position sense, but the application of kinesiology taping does not significantly influence the ability to sense the position of the glenohumeral joint. However, kinesiology taping significantly improves shoulder function in the patient’s subjective assessment. Therefore, the analyzed kinesiology taping application can be used to enhance shoulder function and improve subjective sensations in patients with post-traumatic anterior shoulder instability.

## Figures and Tables

**Figure 1 sensors-25-02326-f001:**
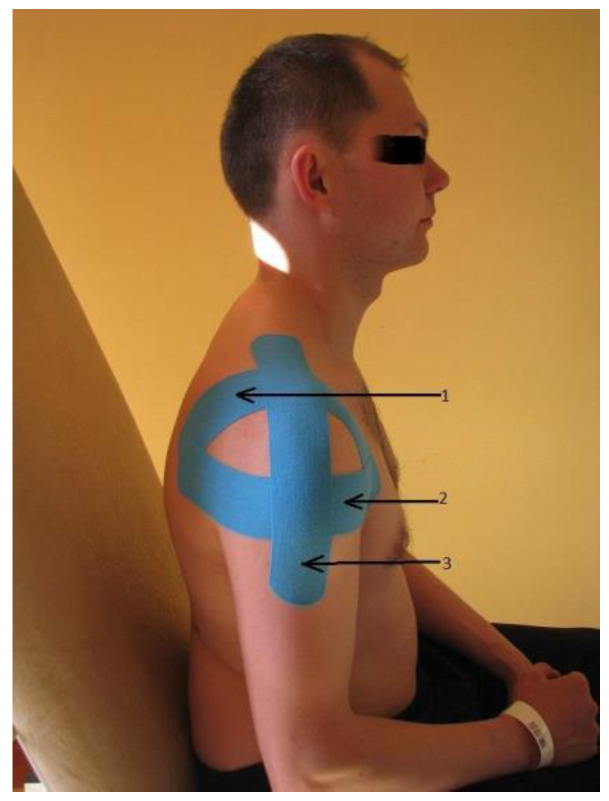
Kinesiology taping application: 1—ligament technique, 2—modified muscular technique, 3—functional technique.

**Figure 2 sensors-25-02326-f002:**
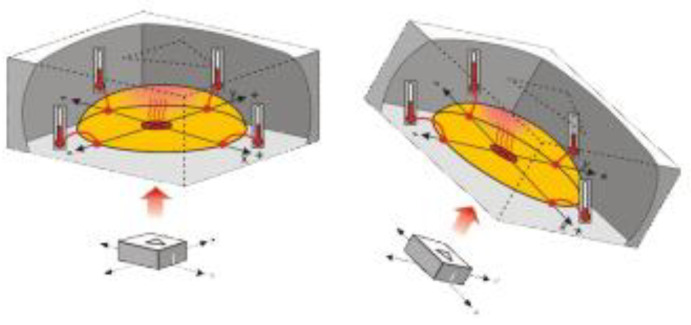
Accelerometer operation diagram. When the accelerometer is level, the hot gas rises to the top center of the chamber, causing all sensors to measure the same temperature. Tilting the accelerometer shifts the hot gas, allowing detection of static and dynamic acceleration by comparing the sensor temperatures, with the accelerometer converting these into pulse duration signals for microcontrollers.

**Figure 3 sensors-25-02326-f003:**
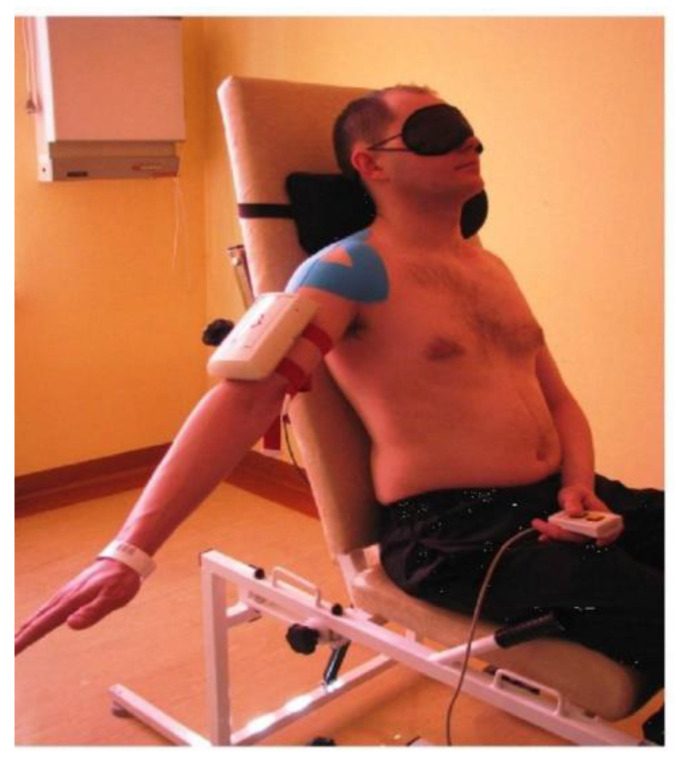
Patient’s position during joint position sense examination with the position-sensing transducer fixed to the arm.

**Figure 4 sensors-25-02326-f004:**
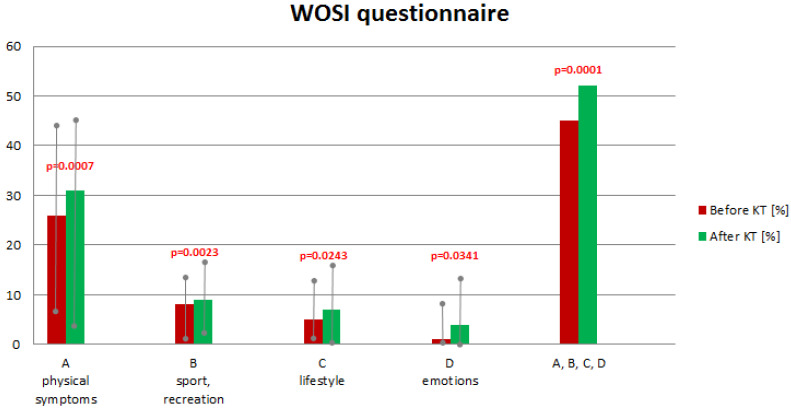
Comparison of the overall result and the average results and range in individual groups of questions of the WOSI survey before and after the use of KT.

**Table 1 sensors-25-02326-t001:** Comparison of the EARJP results of the examined patients with an unstable shoulder before and after the use of KT.

Direction of Motion and Angle [°]		EARJP [°]		*p*
		Before KT	After KT	
		(Standard Deviation [°])	(Standard Deviation [°])	
Abduction	60	6.27 (3.67)	6.62 (3.59)	* 0.5857
	90	4.92 (2.57)	4.58 (2.42)	* 0.6509
	120	4.13 (2.34)	3.78 (1.66)	0.3498
Flexion	60	6.87 (4.05)	6.29 (3.49)	* 0.5104
	90	3.73 (2.26)	3.69 (2.10)	* 0.7621
	120	3.22 (2.13)	3.71 (1.62)	* 0.0897
Internal rotation	30	4.21 (1.98)	4.67 (2.20)	* 0.4284
	45	3.42 (2.42)	3.06 (1.98)	* 0.2949
	60	2.39 (1.92)	2.59 (1.96)	* 0.5629
External rotation	30	3.18 (2.03)	3.01 (1.74)	* 0.3931
	45	2.10 (1.82)	2.16 (2.07)	* 0.9741
	60	1.06 (1.28)	1.44 (1.81)	* 0.2585

EARJP—error of active reproduction of the joint position, expressed in angular degrees. KT—kinesiology taping. Student’s *t*-test was used for related variables. * The Wilcoxon test was used.

**Table 2 sensors-25-02326-t002:** Comparison of the average EARJP value and statistical significance in the control group for the taped extremity before and after KT.

Direction of Motion and Angle [°]		EARJP [°]		*p*
		Before KT	After KT	
		(Standard Deviation [°])	(Standard Deviation [°])	
Abduction	60	4.42 (2.58)	4.82 (3.39)	* 0.6628
	90	3.05 (1.48)	3.42 (1.69)	* 0.1325
	120	3.08 (1.16)	3.42 (1.51)	* 0.4887
Flexion	60	4.51 (2.64)	5.1 (3.39)	* 0.2853
	90	3.06 (1.30)	2.84 (1.56)	* 0.3471
	120	3.08 (1.21)	3.33 (1.61)	* 0.2152
Internal rotation	30	3 (1.24)	3.23 (1.76)	* 0.7869
	45	2.89 (1.18)	2.72 (0.89)	0.4088
	60	3.19 (1.46)	3.37 (1.60)	* 0.5720
External rotation	30	3.29 (1.53)	3.09 (1.14)	0.4903
	45	3.11 (1.60)	3.01 (1.21)	* 0.7131
	60	3.16 (1.63)	3.31 (1.72)	* 0.7247

EARJP—error of active reconstruction of the joint position, expressed in angular degrees. KT—kinesiology taping. Student’s *t*-test was used for related variables. * The Wilcoxon test was used.

**Table 3 sensors-25-02326-t003:** Comparison of the EARJP results in the unstable joint in the study group with the EARJP results of the control group of the taped limb before and after KT.

Direction of Motion and Angle [°]		*p* EARJP Study Group, Unstable Joint Before KT vs. EARJP Control Group, Taped Joint Before KT	*p* EARJP Study Group, Unstable Joint After KT vs. EARJP Control Group, Taped Joint After KT
Abduction	60	** 0.0506	** 0.0182
	90	** 0.0024	** 0.0395
	120	* 0.0311	** 0.2088
Flexion	60	** 0.0077	** 0.1370
	90	** 0.4732	** 0.0530
	120	** 0.9266	** 0.3141
Internal rotation	30	** 0.0087	** 0.0038
	45	** 0.4532	** 0.7272
	60	** 0.0329	** 0.0546
External rotation	30	** 0.7873	* 0.8198
	45	** 0.0227	** 0.0069
	60	** 0.0001	** 0.0001

EARJP—error of active reproduction of the joint position, expressed in angular degrees. KT—kinesiology taping. * Student’s *t*-test with separate variance estimation was used. ** Mann–Whitney U test was used. Red color indicates statistically significant results.

## Data Availability

Data are contained within the article.

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
