# Peer review of "Using an Electronic Goniometer to Assess the Influence of Single-Application Kinesiology Taping on Unstable Shoulder Proprioception and Function"

_sensors, 2025, doi:10.3390/s25072326_

Round 1

Reviewer 1 Report (Previous Reviewer 1)

Comments and Suggestions for Authors

I would like to thank the authors for the changes they have made. I particularly appreciate that they welcomed my suggestion to change the title and methodology, for better clarity and transparency.

I believe that the article now deserves to be published in this form.

Author Response

Thank you for your positive feedback. All comments are very important to us to improve the quality of our work.

Reviewer 2 Report (New Reviewer)

Comments and Suggestions for Authors

The authors examined the effect of kinesiology taping on joint position sense in anterior, post-traumatic shoulder joint instability. The results are worth publishing, but the authors could improve a few points.

1) In the abstract, the authors say that "Few studies describe the impact of taping on proprioception, mainly focus on the knee joint...". If possible, the authors could discuss how the impact of taping is different for the shoulder from that for the knee.

2) The authors conclude that the propriometer goniometer is useful for assessing the impact of Kinesiology taping on post-traumatic anterior shoulder instability. However, the authors did not test the usefulness of the propriometer in this study. The authors could discuss why they came to that conclusion by comparing the proriometer with alternative methods of measuring joint angle.

3) For tables, they use a comma for the decimal point, but using a period is more common. Please change that.

Author Response

The authors examined the effect of kinesiology taping on joint position sense in anterior, post-traumatic shoulder joint instability. The results are worth publishing, but the authors could improve a few points.

Answer:

Thank you for your positive feedback. All comments are very important to us to improve the quality of our work.

  • In the abstract, the authors say that "Few studies describe the impact of taping on proprioception, mainly focus on the knee joint...". If possible, the authors could discuss how the impact of taping is different for the shoulder from that for the knee.

Answer 1

Thank you for this remark. We have described in more detail in the discussion how taping affects knee proprioception and have added the following sentences:

The study with patellar taping demonstrated that taping modulates brain activity in several areas of the brain during a proprioception knee movement task. This decreased BOLD response in the lateral primary sensory cortex was associated with less activity for the knee proprioception task with taping. [27]. We can olny suppose that similar impact could be observed in MRI examination during shoulder taping. The limited studies on this topic are not sufficient to clearly say the differences and real impact of kinesiotaping on a specific joint.

  • The authors conclude that the propriometer goniometer is useful for assessing the impact of Kinesiology taping on post-traumatic anterior shoulder instability. However, the authors did not test the usefulness of the propriometer in this study. The authors could discuss why they came to that conclusion by comparing the proriometer with alternative methods of measuring joint angle.

Answer 2

In the present study, we did not qualitatively compare the examination using the Propriometer with other devices. This is a very interesting topic for future research. Our device had previously been validated and used in studies on unstable shoulder and elbow. In this study, we examined the possibility of using the Propriometer after taping therapy. However, in the future, we will attempt to conduct studies comparing different devices.

  • For tables, they use a comma for the decimal point, but using a period is more common. Please change that.

Answer 3

The tables have been corrected.

Reviewer 3 Report (New Reviewer)

Comments and Suggestions for Authors

Comments:

This paper assessed the usability of the electronic goniometer for examining the effects of Kinesiology Taping on unstable shoulders, and evaluated the impact of a single application of Kinesiology Taping on joint position sensing and limb function in anterior, post-traumatic shoulder joint instability. Experimental results indicate that Kinesiology Taping does not affect the proprioception of the shoulder joint but improves the subjective sense of stability of the patients’ joint. However, this manuscript is not well organized, I have some major issues and minor issues about this manuscript.

Major concerns

  1. The article title may be too long for readers to get the main idea of this study. Besides, ‘Propriometer’ seems not a word in a dictionary. May need to use a more common word.
  2. Lines 7 to 20, all authors belong to the same institution. Do not need to repeat this information.
  3. In the abstract, two sentences ‘Few studies describe the impact of taping on proprioception, mainly focusing on the knee joint, but there is a lack of research evaluating this relationship in unstable shoulder joints.’ and ‘The current study concerns the assessment of the effects of taping which has not been previously tested with this device.’ may repeat with each other.
  4. The structure of the introduction
  5. In the section of ‘Materials and Methods’, the manufacturer information of the taping should be clearly described. The same issue for “Propriometer”.
  6. In the section of ‘Materials and Methods’, it is said ‘The Propriometer uses the earth's magnetic field to measure the range of motion of the joint and reads its position in space in real-time with an accuracy of 0.1 degrees in all axes.’ How does the “Propriometer” work? Is it a magnetic and inertial sensor? Please clarify this. Are the results affected by the magnetic disturbance introduced by the chair?
  7. In the section of ‘Materials and Methods’, it is said ‘which displays the target joint angle set for the patient by operator and the reproduced shoulder angle by patients.’ But in Figure 2, the subject was blindfolded. How did the subject get the target angle when they reach the target angle? Is there a feedback or a notification from the operator?
  8. In Table 1 and Table 2, what’s the meaning of the value in the bracket? Moreover, the units of these data should be written in the head of the table or the caption.
  9. The tables are all too large. The font size is too big. It would be better to make some figure results rather than only tables.
  10. In this paper, there are lots of sentences you can use the abbreviation as you have already defined abbreviations. Such as Kinesiology Taping (KT) and the Western Ontario Shoulder Instability Index (WOSI). Please check the entire manuscript carefully.
  11. In this paper, there are some places where there is only an average value but no standard deviation about the ages of subjects, which is not consistent (line 88 to line 94).
  12. The style of the conclusion is quite unnormal. One paragraph is OK for a general paper.

Comments on the Quality of English Language

n/a

Round 2

Reviewer 3 Report (New Reviewer)

Comments and Suggestions for Authors

Thank you for the revision, which has improved the quality of the manuscript.
However, my previous comment (Comment 11) was not fully addressed. Specifically, please provide the standard deviation in the bar chart (Fig. 4)

Comments on the Quality of English Language

N/A

Author Response

Comment:

Thank you for the revision, which has improved the quality of the manuscript.
However, my previous comment (Comment 11) was not fully addressed. Specifically, please provide the standard deviation in the bar chart (Fig. 4)

Answer:

We are grateful for all the comments. They have improved the quality of the manuscript. Table 4 has been corrected.

This manuscript is a resubmission of an earlier submission. The following is a list of the peer review reports and author responses from that submission.

Round 1

Reviewer 1 Report

Comments and Suggestions for Authors

Thank you for the opportunity to review this article. The objective of this study was assess the impact of Kinesiology Taping (KT) on joint position sense and limb function, in post-traumatic shoulder instability. I appreciate the research topic, which I believe could be of interest, because of the potential benefits that could be derived from an easily implemented intervention.

I have doubts about the chosen methodology, in particular the very short observation period (3 days) that was chosen by the authors. I believe that the duration of the intervention is not sufficient to be able to measure the effects of the application of KT. Furthermore, the authors' choice not to administer any type of exercise meant that significant improvements in joint position sense were to be expected. Within the limits of the study, I would emphasise this concept, as the effectiveness of the application of KT could be higher if combined with a specific rehabilitation protocol.

Here are my other comment:

·       Abstract, line 24: remove the abbreviation, as it is already made explicit in line 21.

·       Abstract, lines 29-31: remove Propriometer information and insert it in the text.

·       Introduction, line 46: please add more recent bibliographical entries on the topic

·       Introduction, line 51: proprioception deficits can also be improved through exercise, I invite the authors to include information on this topic.

·       Materials and Methods, line 76: please replace “studies” with “measures”.

·       Materials and Methods, line 79: I invite the authors to explain why they chose an observation period of only three days.

·       Materials and Methods, line 87: I invite authors to include the technical details of the instrument used

Reviewer 2 Report

Comments and Suggestions for Authors

This article investigates why kinesiology taping benefits the unstable glenohumeral joint (GHJ), hypothesizing that its positive effects arise from enhanced proprioception. While the concept of improving proprioception through taping to refine movement quality and stabilize the GHJ is widely recognized, limited studies validate this mechanism, leaving it uncertain. In this study, the authors evaluated proprioception using the error of active reproduction of the joint position (EARJP) in different static postures and quantified subjective GHJ stability using the WOSI questionnaire. The results suggest that the beneficial effect is not due to proprioception enhancement but rather an improvement in GHJ stability. However, several shortcomings need to be addressed.

Major Issues

  1. Contradiction with Existing Studies:
    Previous studies have attributed the beneficial effects of kinesiology taping to proprioception enhancement. However, this study presents a different conclusion. How do the authors explain this discrepancy?
  2. Validity of Proprioception Measurement:
    Proprioception is the sensory ability to perceive position and movement in space. Using the EARJP at static postures to measure proprioception might not fully capture this dynamic sensation. How can the authors ensure that this method is a valid measure of proprioception?
  3. Statistical Representation:
    Presenting only the mean values in the statistical analysis may not adequately reflect the variability across all subjects. Providing additional statistical measures, such as standard deviation or interquartile range, would give a more comprehensive picture.
  4. Table 3 Logic:
    The rationale for the comparisons made in Table 3 is unclear. Could the authors clarify the relationships between the parameters? Additionally, comparing the "deduction" (difference between before and after) between the control and experimental groups might better quantify the effect of taping.

Minor Issues

  1. Variation in Unstable GHJ Types:
    The study applies a single taping method to all types of unstable GHJ. Considering the variability in instability types, is this approach appropriate?
  2. Objective Data in Table 4:
    The WOSI questionnaire results indicate a significant improvement in GHJ stability after taping. However, the reliance on subjective data alone is a limitation. Including more objective physical examination data would strengthen the findings.
  3. Control Group Data for WOSI Questionnaire:
    The WOSI questionnaire data was collected only from the experimental group. Why was the control group excluded? Providing data from both groups would allow for a clearer interpretation of the intervention’s effectiveness.

Round 2

Reviewer 1 Report

Comments and Suggestions for Authors

I want to thank the authors for responding to the comments, I believe the changes made have improved the quality of the article.

However, I believe the methodology, particularly concerning the choice of observation time, needs further investigation. Indeed, although it is correct to state that the elastic polymer properties decrease 3-4 days after application, nothing prohibits replacing the previously applied KT with a new one, whose properties would be intact. Consequently, it would be expected that prolonging the treatment by using more applications could give different and/or better results.

However, if the authors aimed to evaluate only the effects of a SINGLE APPLICATION of KT, I suggest specifying this (in the title and in the methodology described) as it is a very important aspect for the readers.

Author Response

Comment:

I want to thank the authors for responding to the comments, I believe the changes made have improved the quality of the article.

However, I believe the methodology, particularly concerning the choice of observation time, needs further investigation. Indeed, although it is correct to state that the elastic polymer properties decrease 3-4 days after application, nothing prohibits replacing the previously applied KT with a new one, whose properties would be intact. Consequently, it would be expected that prolonging the treatment by using more applications could give different and/or better results.

However, if the authors aimed to evaluate only the effects of a SINGLE APPLICATION of KT, I suggest specifying this (in the title and in the methodology described) as it is a very important aspect for the readers.

Response:

Thank you for your opinion. We agree that by specifying that we used a single application of KT our article will be clearer, so we changed the title, abstract and methodology section.

Reviewer 2 Report

Comments and Suggestions for Authors

We appreciate the authors' efforts to enhance the clarity of data representation by including standard deviations. However, there remain some unresolved questions that prevent us from providing further constructive feedback at this stage. We regret that we are unable to assist in advancing the quality of the article further and wish the authors success in their future revisions.

Author Response

Comment:

We appreciate the authors' efforts to enhance the clarity of data representation by including standard deviations. However, there remain some unresolved questions that prevent us from providing further constructive feedback at this stage. We regret that we are unable to assist in advancing the quality of the article further and wish the authors success in their future revisions.

Response:

Thank you for your opinion. We did our best to improve our article, and we are ready to correct the indicated points.